# Tourism, Transportation and Low-Carbon City System Coupling Coordination Degree: A Case Study in Chongqing Municipality, China

**DOI:** 10.3390/ijerph17030792

**Published:** 2020-01-28

**Authors:** Fumin Deng, Yuan Fang, Lin Xu, Zhi Li

**Affiliations:** 1Business School, Sichuan University, Chengdu 610065, China; dengfm@scu.edu.cn (F.D.); fangyuan@stu.scu.edu.cn (Y.F.); xulin0329@stu.scu.edu.cn (L.X.); 2The Economy and Enterprise Development Institute, Sichuan University, Chengdu 610065, China

**Keywords:** coupling coordination degree model (CCDM), tourism, transportation, low-carbon city, entropy weight method, gray relational analysis

## Abstract

Tourism and transportation have extremely complex interactions. Tourism developments have expanded demand and stimulated transportation development, which has consequently affected the environment of cities striving towards low-carbon sustainable development. Therefore, there has been an increased research focus on the coordinated binary development of the tourism and transportation industries to ensure sustainable low-carbon cities. To this end also this paper first developed a comprehensive evaluation index system with three subsystems, seven aspects, and 31 indicators. Then, entropy weight and gray correlation were combined to determine the index weights and a physics coupling concept employed to build a tourism, transportation and low-carbon city (TTLC) coupling coordination degree model, which was then applied to quantitatively analyze the coupling and evolutionary trends in Chongqing’s TTLC efforts from 2008 to 2017. It was found that the overall coupling coordination was volatile and rising, and that industry scale, industry performance, and energy consumption had the most significant impact on the coupled systems, indicating that these key factors must be considered in macro decision-making. In general, it was shown that the combination of the coupling coordination degree model and entropy weight gray correlation was able to effectively evaluate dynamic coupling relationships.

## 1. Introduction

As China has developed, more people have moved into the middle class, dispose of higher incomes and have more leisure time. Consequently, the tourism industry has become an important driving force in China’s continued economic development. The Ministry of Culture and Tourism reported that China’s tourism industry contributed 9.94 trillion CNY or 11.04% to Chinese GDP in 2018. Tourism is also contributing to regional economic development and urbanization [1] and has become closely associated with the transportation industry. An important foundation for the effective development of tourism resources and tourist destinations and an important indicator of national regional and local tourism advancement is the availability of efficient, convenient transportation networks [2]. At the same time, tourism developments motivate national and regional tourism transportation network improvements.

However, these tourism and transport developments have had an adverse effect on the ecological environment [3] by exacerbating traffic jams, and increasing noise, waste, water and air pollution. In recent years, there has been a move toward low-carbon economies, societies and cities. While cities are the main tourism areas, they are also the main generators of greenhouse gas emissions such as carbon dioxide. Studies by the International Energy Council (IEA, 2008) found that cities were responsible for 70–75% of global energy consumption and 80% of tritium CO_2_ emissions. There has been a global consensus on climate change and a binding intergovernmental agreement reached to reduce carbon dioxide emissions. Consequently, the Chinese government has proposed a green, low-carbon, cyclical development path with the aim that by 2030, the CO_2_ emissions per unit of GDP will only be 60–65% of 2005 levels. However, to achieve this aim, urban carbon emissions must be reduced and urban low-carbon sustainable economies developed. Therefore, coupling and coordinating tourism, transportation and low carbon (TTLC) is key to achieving sustainable economic and social development.

Therefore, to provide a new explanation for sustainable TTLC development theory and with the overall aim of expanding the current research scope and constructing a three-in-one interactive tourism-transportation-low-carbon development model, this paper examined the interactions between tourism, transportation and low-carbon in tourism cities, and as an example, empirically analyzed the coupling and coordination of these three systems in the Chongqing municipality in central and western China. The results from this study give insights into low-carbon tourism development and transportation planning in China, and provide a realistic policy basis for the governance of urban carbon emissions and the sustainable development of TTLC environments.

## 2. Literature Review

There are many contradictory relationships between TTLC environments because of the tourism industry’s environmental dependence and the transportation industry’s resource consumption and emissions. As few studies have explored the complex relationships between these three subsystems, it is necessary to investigate the separate relationships between tourism and transportation, tourism and low-carbon cities, and transportation.

### 2.1. Relationship between Tourism and Transportation

Foreign scholars have been studying the relationships between tourism and transportation developments from the early 20th century. Generally speaking, tourism and transportation have been found to both promote and restrict each other. Tourism transport system operations and transport mode interactions support tourism passenger and freight flows in and out of destinations and provide transport within the destination and connecting transport services within the region [4]. As the provision of various transportation modes such as air, railway, bus, and other tourist transportation assists in the development of national and international tourism [5], transportation network improvements play a vital role in the sustainable tourism development [6,7]. Taking Yunnan as an example, Xi et al. analyzed the influences on regional tourism spatial structure changes [8], and Becken et al. found the transportation link energy consumption was the largest of all tourism activity links. Therefore, it has become essential to find ways to reduce transportation link energy consumption, improve resource utilization efficiency, and promote sustainable tourism industry development. While there has been some research focused on the role of transportation in the development of tourism, few studies have examined the adverse effects of transportation on tourism development [9]. Khadaroo et al. used a gravity framework and found that transportation infrastructure was an important determinant of destination tourist flows, and that the sensitivity of these tourist flows to transportation facilities varied based on passenger source and destination [10]. 

### 2.2. Relationship between Tourism and Low-Carbon Cities

Because tourists either visit cities or transit through cities, cities could be seen to be tourism carriers; therefore, there have been several studies on the relationships between cities and tourism. When studying the relationship between tourism and cities, more consideration should be given to the ecological environment and socio-economic factors. Mullins believed that tourism played a significant role in promoting urbanization and was the first to propose an “urban urbanization” concept in 1991 [11]. Deng et al. conducted an empirical analysis of tourism-dependent cities, and concluded that the only way to avoid a “resource curse” in tourist cities and ensure the prosperity of the tourism industry was economic diversification [12]. Shu et al. analyzed the coupling and coordination relationships between the tourism industry and an ecologically civilized city from the four aspects of the economy, culture, society, and ecology, and exemplified Guiyang to demonstrate the feasibility of the coupling and coordination model [13].

When the “low-carbon economy” was proposed, low-carbon considerations began to be included in research on cities and tourism. Cai et al. believed that low-carbon cities were an advanced stage of livable urban ecological development, which has become a common urban development pursuit around the world, and also a strategic choice for the transformation of China’s tourist cities [14]. For example, Xiao et al. combined low-carbon city construction with global tourism development, and developed a low-carbon city system based on the five major elements of population, economy, society, resources, and environment, after which a system dynamics method was employed to dynamically reflect the urban operational mode and the system element interactions [15], and Zhang et al. established a system dynamics model to explore the evolutionary characteristics of urban low-carbon tourism systems under different scenarios, and found that under the economic priority scenario, tourism development contributed to increased pollution levels, but the risk was manageable over the long run in low-carbon tourism systems [16].

### 2.3. Relationship between Transportation and Low-Carbon Cities

More than 50% of the world’s population live in urban environments and 75% of total global carbon emissions are generated in cities; therefore, cities are the main focus for energy conservation and emissions reduction efforts [17]. The urban emissions from industry, transportation and construction account for 40.4% of total global greenhouse gas emissions [18], and as the number of urban motorized travel tools increases, the carbon emissions commensurately increase; therefore, the development of low-carbon public transportation system has become a global priority.

Research on the relationship between the transportation industry and low-carbon cities has tended to emphasize the importance of low-carbon city transportation. For example, Zhang et al. believed that the transportation industry focus should be on green, low-carbon transportation [19], Jia concluded that urban road traffic management policies heavily influenced the development of low-carbon cities, and proposed several urban road low-carbon city construction traffic planning and management policies [20], Ye et al. proposed an improved green transportation-oriented system development model based on the belief that optimized urban spatial structures and low-carbon travel played very important roles in promoting sustainable low-carbon city transportation and development [21], and Matsuhashi et al. studied the potential of passenger vehicles to reduce carbon dioxide emissions over the long run, introduced a compact city concept, and based on this concept, conducted a passenger car carbon dioxide emissions scenario analysis for 2030 under both compact and decentralized scenarios [22].

Based on previous research, this paper studied the relationships between tourism, transportation, and low-carbon cities (Figure 1), and taking Chongqing as an example, established an TTLC evaluation index system based on 2008 to 2017 data, and then used entropy weight and gray correlation coupling coordination methods to determine the optimal weights and the comprehensive system evaluation indicator development levels. The degree model was then employed to dynamically and quantitatively analyze the coupling and coordination relationships between the three subsystems and formulate TTLC development coordinated development strategies for the relevant Chongqing departments. Therefore, the innovations of this research are as follows.

While there have been many studies on the relationship between system pairs, there have been few studies on the coordinated development of the three tourism, transportation and low-carbon city subsystems. Further, even though there has been comprehensive research on the relationships between industry and low-carbon cities, there has been less focus on the coupling relationships; therefore, this paper extends previous results and adds to the research discipline.

A comprehensive TTLC evaluation index system is developed that has three subsystems, seven aspects, and 31 indicators to ensure a scientific and objective evaluation.

Previous coupled quantification models have partially relied on subjective quantification methods, and while these can make the empirical results easier to interpret, the veracity of the results can be affected because different experts have different subjective views on the related issues. However, in this paper, gray correlation analysis and the entropy weight method are combined before the coupling and coordination analysis to effectively eliminate any bias resulting from the subjective factors in the weight calculation, thereby ensuring better results objectivity.

## 3. Study Area 

Chongqing is located on the eastern edge of Sichuan Basin (Figure 2) in the transition zone between the Qinghai-Tibet Plateau and the middle and lower reaches of the Yangtze River. The area under the Chongqing jurisdiction is 470 km wide from east to west and 450 km long from north to south; a total area of 8.24 square kilometers. Chongqing, which is an industrial, commercial, land, water and air transportation hub in southwest China, is divided into 38 districts and counties, is under the unified management of the central government, is the largest economic center in the upper reaches of the Yangtze River, connects the developed areas in the east and the west, and has a superior geographical location.

Chongqing is an important strategic fulcrum for western regional development. At the junction of the “Belt and Road” and the Yangtze River Economic Belt, Chongqing has been playing a unique and important role in China’s regional opening up and development. Chongqing is currently going through a critical economic development period. Because of the Chongqing’s rapid urban and rural economic development and the consequent rise in environmental pollution and CO_2_ emissions, there has been an increased focus on environmental protection and emissions reduction, with the city being selected as one of the first low-carbon pilot cities in China. Chongqing’s tourism market has also developed, with 59.273171 million domestic and foreign tourists and a total tourism revenue of 434.415 billion CNY in 2018, respective year-on-year increases of 10.13% and 31.32%. Chongqing’s transportation industry has actively responded to its strategic layout and giving full play to its political and location advantages has focused on the strategic goal of developing Chongqing to be a “south-west comprehensive transportation hub”. To do this, it has been vigorously expanding transportation supply, improving the transportation facility network, promoting green transportation development, and developing an integrated, comprehensive transportation system that is smooth, efficient, convenient, green, and well connected. By 2018, the city’s total railway mileage had increased to 2300 km, the total highway mileage had increased to 3100 km, and the transportation capacity had been further enhanced.

## 4. Materials and Methods

### 4.1. Index System

Based on the previous studies [23,24,25,26,27,28,29,30,31,32,33,34,35,36,37,38,39] and scientific, systematic, hierarchical and data availability principles, this paper sought to analyze the coupling coordinated development mechanism between tourism, transportation, and a low-carbon city. Based on the actual situation in Chongqing, representative indicators were selected that reflected TTLC development and the mutual influences between the three subsystems, from which a comprehensive evaluation index system was developed.

The industry scale reflects the overall scale and development level of the industry. The industry scale of the tourism industry includes the total star-rated hotels, total travel agencies, total number of beds in hotels, number of A-grade tourist attractions, employees in the tourism industry, original value of the tourism enterprise fixed assets. The industrial scale of the transportation industry includes length of highways in operation, length of railways in operation, number of civilian passenger cars, municipal area of paved roads. The performance of the tourism industry reflects the capacity of reception and the economic benefits it generates, including international tourism receipts, number of international tourists, the average daily per capita expenditure by international tourists, room occupancy rate, tourism operating receipts. The performance of the transportation industry reflects the transportation capacity of various modes of transportation, including railway passenger traffic, highway passenger traffic, civil aviation passenger traffic, railway passenger turnover, highway passenger turnover, civil aviation passenger turnover.

Low-carbon cities aim to implement a low-carbon economy, including low-carbon production and low-carbon consumption, and to establish a resource-saving and environment-friendly society. This paper’s low-carbon city subsystem is developed from three aspects: energy consumption, carbon emissions and low-carbon environment. Energy consumption reflects the consumption and utilization efficiency of urban resources and energy, including total energy consumption, energy consumption per unit of GDP, annual per capita energy consumption, energy consumption elasticity ratio. Carbon emissions reflect the city’s carbon emissions, including three indicators of carbon emissions per capita, total carbon emissions and carbon productivity. The low-carbon environment reflects the quality of urban forest resources and the treatment of environmental pollution, including the innocuous disposal rate of living garbage, afforestation area per capita, urban green coverage rate. Finally, the comprehensive indicator system consists of 3 subsystems, 7 aspects, and 31 indicators (Table 1).

### 4.2. Data Collection and Pre-Processing

The required data were collected from the China Tourism Statistical Yearbook [40], the China Energy Statistical Yearbook [41] and the Chongqing statistical yearbooks [42]. The data were standardized using Formulas (1) and (2) to eliminate the dimension, magnitude, and positive and negative orientation influences [43]. 

For the positive index: (1)$$p_{ij}=\frac{X_{ij}-\underset{1\leq j\leq n}{\min}X_{ij}}{\underset{1\leq j\leq n}{\max}X_{ij}-\underset{1\leq j\leq n}{\min}X_{ij}}$$

For the negative index:(2)$$p_{ij}=\frac{\underset{1\leq j\leq n}{\max}X_{ij}-X_{ij}}{\underset{1\leq j\leq n}{\max}X_{ij}-\underset{1\leq j\leq n}{\min}X_{ij}}$$
where $X_{ij}$ was the standardized value for index *j* in year *i*, and $\underset{1\leq j\leq n}{\max}X_{ij}$ and $\underset{1\leq j\leq n}{\min}X_{ij}$ were maximum and minimum value for index *j* in all years.

### 4.3. Entropy Weight Method

The entropy weight method (EWM), which is based on Shannon entropy [44], is able to measure the relative intensities of contrasting criteria to determine the average intrinsic information for decision-making [45], and is an objective method for constructing judgment matrices based on the evaluation index value and determining the associated weights based on the degree of variation in each index. When the weight of each principal component is determined using the entropy method, the influences of the subjective factors can be significantly reduced, providing better, more practical results. The steps taken to determine the entropy weight are as follows:

(1) Standardization of principal component indexes.

With *n* measured objects (city) and m principal component factors, the standardization matrix established based on each principal component score for the measured object is $R={\left(p_{ij}\right)}_{n\times m}\left(i=1,2,\ldots ,n;j=1,2,\ldots ,m\right)$. 

(2) Determination of entropy and entropy weight of principal component factors. 

In line with the entropy definition, the entropy value $e_{j}$ and entropy weight $r_{j}$ of the j-th principal component factor are:(3)$$e_{j}=-k\overset{m}{\underset{i=1}{\sum }}\left[p_{ij}\ln p_{ij}\right]$$
where *k* is Boltzman’s constant (1/lnm), which guarantees that $0\leq e_{j}\leq 1$.

(3) The diversification degree $\bar{e_{j}}$ of the information provided by the alternative evaluation value for criteria *j* is defined as $\bar{e_{j}}=1-e_{j}$.

(4) Criteria normalization. 

To satisfy the GRA limitations, the relative weights are within the range (0,1), with the value obtained in Equation (3) being normalized using Equation (4): (4)$$r_{j}=\frac{\bar{e_{j}}}{\sum_{j=1}^{n}\bar{e_{j}}}$$
where $\overset{n}{\underset{j=1}{\sum }}r_{j}=1$.

### 4.4. Gray Relational Analysis

A gray system solves problems that have discrete data, partial information and ambiguity. The distinguishing feature of gray system theory is that it can handle smaller data easily and while it does not seek to find the best solution, it provides good solutions to real world problems [46].

The data collected were alternatives ($x_{i}$), evaluation criteria ($c_{j}$), relative weights ($w_{j}$), and alternative performances under each criteria ($x_{ij}\left(j\right)$), all of which were normalized in advance. The aspired ($x^{*}$) and the worst ($x^{-}$) values for the alternatives were then identified using a performance matrix, where $x^{*}\left(j\right)$ = $\max_{i}\left(x_{ij}\left(j\right)\right)$ and $x^{-}\left(j\right)$ = $\min_{i}\left(x_{ij}\left(j\right)\right)$.

The GRA analysis had the following steps:

(1) Gray relation coefficients for the aspired values:(5)$$\gamma \left(x^{*}\left(j\right),x_{i}\left(j\right)\right)=\frac{\underset{i}{\min}\;\underset{j}{\min}\left|x^{*}\left(j\right)-x_{i}\left(j\right)\right|+\zeta \underset{i}{\max}\;\underset{j}{\max}\left|x^{*}\left(j\right)-x_{i}\left(j\right)\right|}{\left|x^{*}\left(j\right)-x_{i}\left(j\right)\right|+\zeta \underset{i}{\max}\;\underset{j}{\max}\left|x^{*}\left(j\right)-x_{i}\left(j\right)\right|}$$

Gray relation grade (larger is better):(6)$$\gamma \left(x^{*},x_{i}\right)=\overset{n}{\underset{j=1}{\sum }}w_{j}\gamma \left(x^{*}\left(j\right),x_{i}\left(j\right)\right)$$
where the weight $w_{j}$ is obtained by entropy.

(2) Gray relation coefficients for the worst values:(7)$$\left(x^{-}\left(j\right),x_{i}\left(j\right)\right)=\frac{\underset{i}{\min}\;\underset{j}{\min}\left|x^{-}\left(j\right)-x_{i}\left(j\right)\right|+\zeta \underset{i}{\max}\;\underset{j}{\max}\left|x^{-}\left(j\right)-x_{i}\left(j\right)\right|}{\left|x^{-}\left(j\right)-x_{i}\left(j\right)\right|+\zeta \underset{i}{\max}\;\underset{j}{\max}\left|x^{-}\left(j\right)-x_{i}\left(j\right)\right|}$$

Gray relation grade (larger is worse, smaller is better):(8)$$\gamma \left(x^{-},x_{i}\right)=\overset{n}{\underset{j=1}{\sum }}w_{j}\gamma \left(x^{-}\left(j\right),x_{i}\left(j\right)\right)$$

(3) Relative gray relation scores. Combine (3) above for the ranking based on the relative gray relationships for the aspired and worst values:(9)$$R_{i}=\frac{\gamma \left(x^{*},x_{i}\right)}{\gamma \left(x^{-},x_{i}\right)}$$

### 4.5. Coupling Coordination Degree Model (CCDM)

Coupling, a concept from physics, refers to the influences that two systems or forms of motion have on each other through their various interactions [35], with the coupling degree being the degree of interaction and influence within a system or between the system elements [47]. Coupling coordination degree models (CCDM) have been widely used in the nature, economic and societal fields. The relationship between tourism, transportation and low-carbon urban development can be seen as an effect feedback mechanism; that is, a coupled system consisting of a tourism subsystem, a transportation subsystem and an LCC subsystem. The general form for the coupling degree of these three subsystems is shown in Equation (10) [48]:(10)$$C={\left\{\frac{U_{1}U_{2}U_{3}}{{\left[\frac{U_{1}+U_{2}+U_{3}}{3}\right]}^{3}}\right\}}^{\frac{1}{3}}$$
where C is the coupling degree between tourism, transportation and low-carbon cities, C∈[0,1]. If the value for C is closer to 1, there is a stronger interaction between the three subsystems. $U_{1}$, $U_{2}$, $U_{3}$ are the respective comprehensive development levels for the tourism, transportation industry, and low-carbon urban subsystems; that is, the comprehensive evaluation level.

As the coupling degree only reflects the strength of the interactions between the systems rather than the level of system integration, the coordination degree, which measures the degree of harmony between systems or internal elements, is used to reflect the system trends from disorder to order. Therefore, to truly reflect the synergy and coordination degree of the interactions between the three tourism-transportation-low carbon city subsystems, the following coupling coordination degree model is constructed:(11)$$D=\sqrt{C\cdot T}$$
(12)$$T=\alpha U_{1}+\beta U_{2}+\gamma U_{3}$$
where D represents the coupling coordination degree, T is the comprehensive evaluation index for the tourism, transportation and the LCC subsystems, and reflects the overall benefit, and α, β and γ are the respective contributions of each subsystem, α+β+γ=1. As this study assumed that each subsystem was equally important to the coordinated tourism, transportation and low-carbon city development, $\alpha =\beta =\gamma =\frac{1}{3}$. The level of their coupling coordination degree is presented in Table 2.

After calculating the coupling coordination degree for the tourism, transportation and low-carbon city subsystems, the coupling coordination degree has been traditionally and subjectively divided into several levels [49]. However, here, and in reference to previous research, the coupling coordination degree was divided to more objectively reflect the coordinated development levels of the tourism-transportation-low carbon city subsystems [25]. 

## 5. Results

### 5.1. Indicator Weights

The entropy weight method was used to calculate the indicator weights in Chongqing’s tourism, transportation, and low-carbon city subsystems (Table 3). The overall industry scale and industry performance development in the tourism subsystem was found to be relatively balanced, with the number of A-grade tourist attractions (x_4_), the average daily per capita expenditure by international tourists (x_9_), and the room occupancy rate (x_10_) adding up to 38.7%, which indicated that these three indicators were important factors in the tourism subsystem. In the transportation industry subsystem, the industrial performance level was found to be more important than the industrial scale level, which indicated that industrial performance in the transportation industry was a powerful driving force for transportation industry development. The largest secondary indicator contributions accounting for 46.42% were length of railway operations (y_2_), railway passenger traffic (y_5_), highway passenger traffic (y_6_), and railway passenger turnover (y_8_). In the low-carbon city subsystem, the energy consumption weight was 49.87%, indicating that energy consumption was a core issue in the low-carbon city subsystem, followed by carbon emissions and a low-carbon environment, with total energy consumption (z_1_), annual per capita energy consumption (z_3_), and carbon productivity (z_7_) having the greatest impact on low-carbon cities at 46.43%.

### 5.2. Results on the Overall Level of the Subsystems

Using the gray relational analysis model, the comprehensive development levels of Chongqing’s tourism, transportation and low-carbon city subsystems from 2008 to 2017 were calculated. Chongqing’s tourism subsystem comprehensive development rose from 2008 to 2017 (Figure 3). In 2008 and 2009, it was at the lowest levels at 0.187 and 0.156; however, over the subsequent seven years, it slowly rose. The Chongqing municipal government began vigorously developing the tourism industry in 2011 by building a tourism platform and promoting the transformation and upgrading of the tourism industry. In 2017, the tourism industry scale and performance had the most obvious increase and the overall tourism subsystem development also reached its peak, rising from 0.252 in 2016 to 0.550. It can also be seen in Figure 3 that from 2008–2017, the industry scale and comprehensive tourism industry level were consistent, which indicated that tourism industry scale had the most significant impact on tourism subsystem development.

Chongqing’s comprehensive transportation subsystem development (Figure 4) had no obvious changes from 2008 to 2012 and remained at about 0.2. From 2012 to 2013, it rose slowly from 0.207 to 0.270, and in the following two years showed a slight downward trend; however, in the three years following 2015, it rose sharply, from 0.254 to 0.799, reaching its peak. Figure 4 also shows that the industrial performance level curve and the transportation subsystem comprehensive level curve are roughly similar. The industrial scale level curve did not change significantly between 2008 and 2015, and it rose sharply after 2015. From 2015, Chongqing significantly increased its transportation investment in such areas as improved transportation routes and increased road area, which enhanced the transportation industry in this region.

Chongqing’s comprehensive low-carbon city subsystem horizontal curve (Figure 5) was U-shaped, with the lowest points being in 2012 and 2014 at 0.156. From 2008 to 2010, it fell from 0.476 to 0.201 and except for slight fluctuations in 2013, there was a slow decline in the following four years. At this time, the carbon emissions and the low-carbon city comprehensive subsystem trends were similar, indicating that carbon emissions was having a significant impact on the effective construction of a low-carbon city. Excessive carbon emissions will not only cause the greenhouse effect, but also lead to a series of environmental and social problems, thereby restricting the development of low-carbon cities. Since Chongqing was nominated as a low-carbon pilot city in 2010, low-carbon urbanization has been progressing steadily and has had remarkable results. Since 2014, the comprehensive low-carbon city subsystem level had a clear upward trend from 0.156 to 0.374.

### 5.3. Coupling Coordination Degree

The coupling coordination degree between the Chongqing tourism, transportation and low-carbon city subsystems was calculated from 2008 to 2017 using the coupling coordination degree model and the average TTLC system coupling coordination level was found to be 0.51, indicating a reluctant coordination level. As can be seen in Figure 6, the coupling coordination degree had a slight fluctuation at first and then rose, which was most similar to the overall transportation subsystem curve. The comprehensive tourism and low-carbon city subsystem development levels were 0.26 and had similar development trends. Based on the coupling coordination degree criterion, the coupling coordination level was analyzed in three stages, each of which is discussed in the following.

The first phase was from 2008–2014. During this time, the overall TTLC system coupling coordination level decreased slightly from reluctant coordination to approaching imbalance as it was in a transitional development stage. From 2008 and 2009, the low-carbon city subsystem was significantly higher than the comprehensive tourism subsystem and transportation subsystem level. Chongqing has abundant tourism resources, but most of them are untapped. In addition, the rugged terrain and inconvenient transportation environment have made tourists unwilling to travel to Chongqing, which greatly hindered the development of tourism. Although Chongqing’s tourism and transportation industries are lagging behind at this time, they are more environmentally friendly. Correspondingly, the lower energy consumption and lower negative environmental impacts resulted in a higher low-carbon city level. However, overall, the TTLC system development was uncoordinated, and there was a serious development lag in the tourism and transportation industries. From 2011, the Chongqing government began to promote Chongqing tourism across the country and developed the tourism slogan “Chongqing, must go”, which led to an increase in the tourism development level. At the same time, the government began to invest in the transportation industry. Consequently, effective tourism market promotion and convenient transportation promote the continuous development of Chongqing’s tourism industry. However, the development of tourism at this time was relatively extensive, and the flock of tourists caused huge traffic pressure, which inevitably caused a certain impact on the environment. Therefore, the comprehensive low-carbon city subsystem development level continued to decline, and the TTLC system coupling coordination was approaching imbalance.

The second phase was from 2014–2016. The TTLC system coupling coordination level gradually became more coordinated, with the coupling coordination becoming akin to reluctant coordination. At this stage, the government was vigorously developing the tourism industry and continuing to improve the transportation network by expanding capital investment. The increasingly developed transportation system has provided good infrastructure conditions for tourism, and the rapid development of tourism has in turn promoted the improvement of the transportation network. All of which brought Chongqing’s tourism and transportation industry development to a new level. However, the continuous tourism and transportation developments had adverse impacts on the environment and ecology: increased water and soil pollution, excessive energy consumption, rising emissions, and a decline in environmental quality: clearly indicating that more care needs to be taken by tourism planners and government organizations to ensure that there is a coordinated relationship between tourism, transportation and the low-carbon city developments. As one of the first low-carbon pilot cities, Chongqing has implemented many low-carbon city initiatives such as afforestation, urban greening, innocuous waste disposal, and the use of clean energy. In 2015, Chongqing’s per capita carbon emissions decreased by 3% over the previous year, and along with its low-carbon urbanization, Chongqing has transformed and upgraded the tourism industry and strongly promoted the comprehensive, coordinated and sustainable development of the tourism, transportation and low-carbon city systems. In 2015, the low-carbon city subsystem comprehensive development level growth rates doubled from 2014.

The third phase was from 2016–2017. The TTLC system coupling coordination degree was intermediate and had a good development trend. As the social economy developed, people’s living standards improved and more people had the ability and willingness to spend money on traveling. The Chongqing Municipal Government had put significant effort into developing the low carbon tourism industry, advocated rational use of resources, strengthened ecological and environmental protection in tourism areas, and promoted civilized tourism, which accelerated the development of Chongqing tourism toward a healthy direction. Chongqing had gradually become one of the most popular Chinese tourist destinations. 

In 2017, Chongqing was one of the top 50 most popular tourist cities in China in 2017, receiving 542 million domestic and foreign tourists, and earning 330.804 billion CNY in total tourism revenue, year-on-year increases of 20.3% and 25.1%. Safe, convenient and comfortable means of transportation are the first choice for tourists to travel. Increasing tourism demand not only promotes the continuous improvement of transportation infrastructure and services, but also drives the growth of the transportation economy. The comprehensive transportation subsystem level consequently increased rapidly from 0.43 in 2016 to 0.80 in 2017. The comprehensive low-carbon city subsystem level also increased, all of which indicated that Chongqing’s tourism, transportation and low-carbon city construction had well-coordinated development trends. However, it is worth noting that the comprehensive low-carbon city subsystem development rate has slowed down from 2006 to 2017, so policymakers should pay more attention to the development of low-carbon tourism and low-carbon transportation.

These results indicated that the proposed coupling coordination model could be of great assistance to future city development planning. As the comprehensive transportation hub city in the southwestern region and an economic center in the upper reaches of the Yangtze River, Chongqing has rich tourism resources and inherent geographical advantages. Therefore, studying the coordinated relationship between tourism, transportation and low-carbon city systems is important for the sustainable development of the city.

## 6. Conclusions

Tourism and transportation are important components of low-carbon cities, which are vital to sustainable urban development [27]. With Chongqing as the research object, this paper established a comprehensive evaluation index system and employed entropy weight, gray correlation and a coupling coordination degree model to examine the interactions between and quantitatively evaluate the comprehensive development of Chongqing’s tourism, transportation, and the low-carbon city subsystems and determine the coupling coordination between the TTLC systems.

The low-carbon subsystem comprehensive development from 2008–2017 had a U-shaped curve, while the tourism and transportation industry developments had clear upward trends. From 2015 in particular, Chongqing’s comprehensive development rose rapidly because the tourism and transportation industries were promoting each other. However, much of this extensive growth in Chongqing’s tourism and transportation industry was at the cost of energy consumption, which was causing ecological and environmental problems.

The TTLC system coupling coordination became coordinated after 2015. Chongqing’s TTLC system change trends could be divided into three stages: 2008–2014, 2014–2016 and 2016–2017: over the ten years, and were observed to change from a lag in tourism and transportation to synchronous coordinated TTLC system development.

To sum up, the proposed combination coupling coordination degree and entropy weight gray correlation model in this paper was found to be able to effectively and objectively reflect the relationships between the tourism, transportation and low-carbon city systems. Therefore, this model could be used by policy makers to identify the contributing factors to the development of tourism, transportation, and low-carbon city systems, understand their complex coupling relationships, and implement low-carbon city development strategies. This article provides new ideas for eco-tourism, transportation, and the comprehensive and coordinated development of low-carbon cities, with the results giving rise to the following suggestions for policy makers:(1)Increase capital investment in the tourism industry, rationally develop tourism resources, build tourism destinations, tourism products and infrastructure supporting services, and realize intensive use of tourism resources.(2)Reasonably adjust the spatial layout of economic tourist development, vigorously develop eco-tourism and low-carbon tourism, increase the urban ecological environment and ecological civilization, and seek sustainable urban development.(3)Expand the spatial structure of tourism, further shorten the space-time distance between tourist destinations and source areas, and ensure a comprehensive modern tourism transportation network made up of aviation, railways, highways, waterways and scenic circles.(4)From a low-carbon development perspective, adjust industrial structures, develop clean energy, improve energy consumption structures, implement energy conservation and emissions reductions, and reduce the impact of industrial development on the environment [50].

However, this study had some limitations. First, only energy consumption, carbon emissions and a low-carbon environment were considered in the low-carbon city subsystem as these fit the research theme of this article. However, as the internal relationships within low-carbon city systems are complex, more driving factors need to be integrated into the low-carbon city subsystem to ensure comprehensive and more realistic evaluations. Second, as the urban development process and industrial structure of different regions are inconsistent and there is space and time heterogeneity, a macro analysis of the space-time scale could be conducted and more innovative methods developed to simulate the dynamic coupling relationships between systems. These further developments could provide a valuable reference for the formulation of differentiated sustainable urban development policies.

## Figures and Tables

**Figure 1 ijerph-17-00792-f001:**
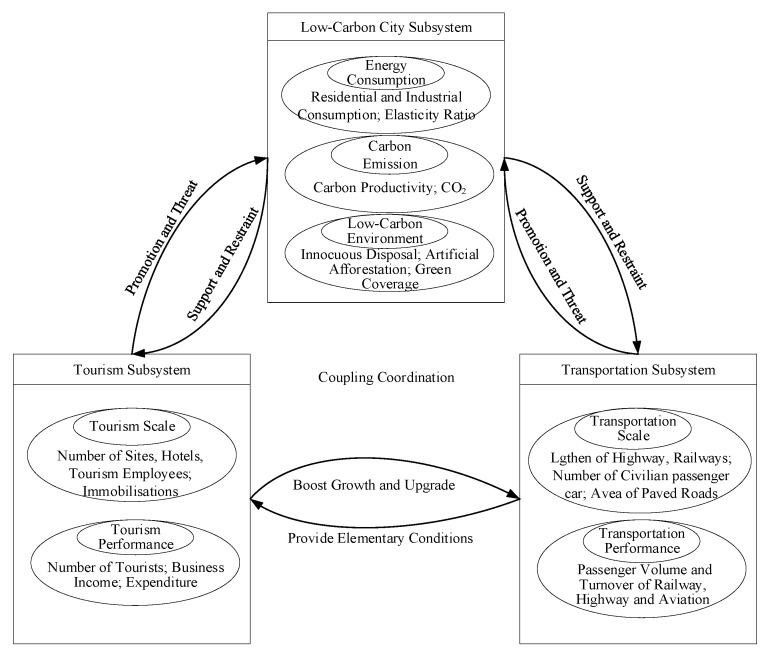
Relationship between tourism, transportation and low-carbon cities.

**Figure 2 ijerph-17-00792-f002:**
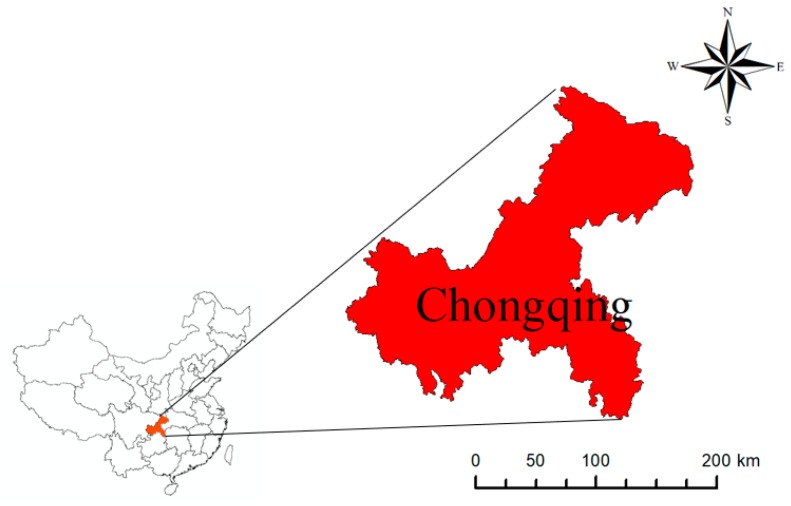
Location of Chongqing municipality in China.

**Figure 3 ijerph-17-00792-f003:**
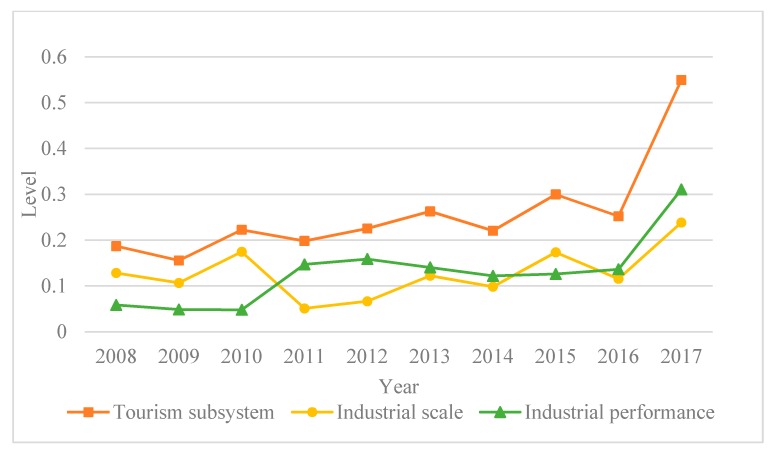
Comprehensive tourism subsystems.

**Figure 4 ijerph-17-00792-f004:**
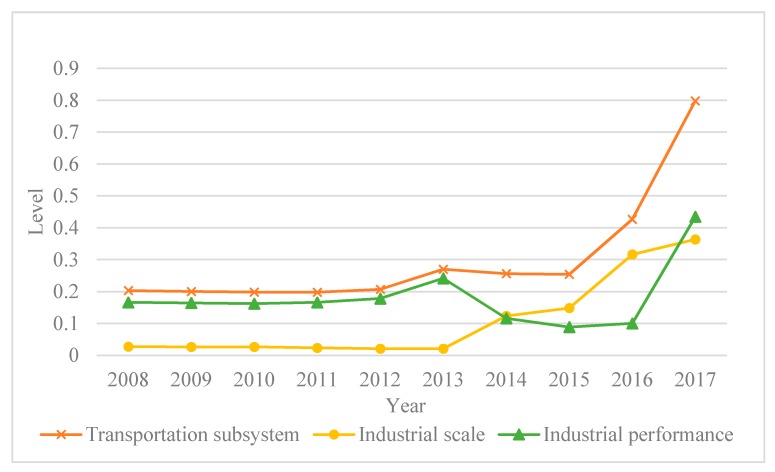
Trends of the comprehensive levels in transportation subsystem.

**Figure 5 ijerph-17-00792-f005:**
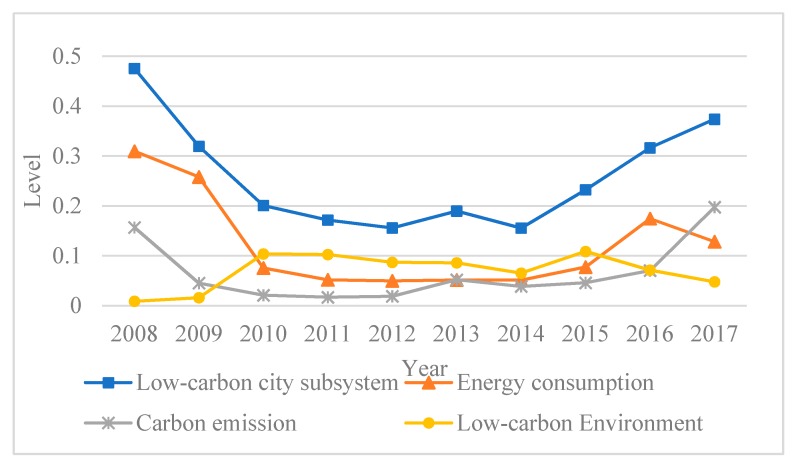
Comprehensive LLC subsystem levels.

**Figure 6 ijerph-17-00792-f006:**
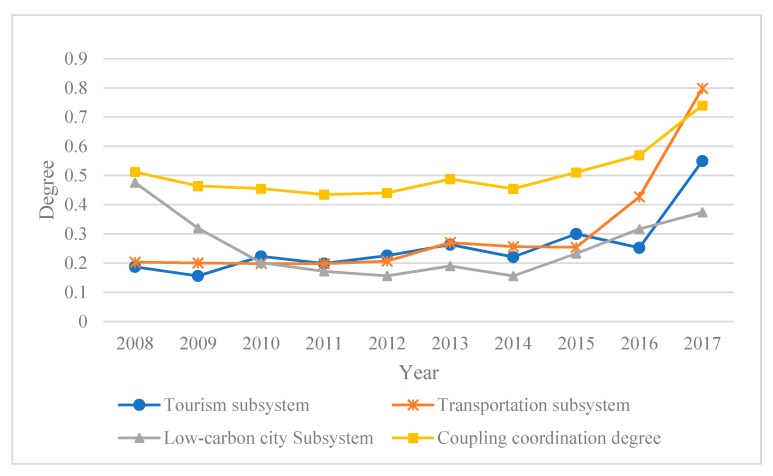
Coupling and coordination degree of Chongqing’s TTLC systems from 2008 to 2017.

**Table 1 ijerph-17-00792-t001:** Comprehensive evaluation index system.

Subsystem	First-Class Index	Second-Class Index	References
Tourism subsystem (x)	Industrial scale	Total star-rated hotels (number) (x_1_)	[26,27,28,30,33,34]
		Total travel agencies (number) (x_2_)	[26,27,29,34]
		Total number of beds in hotels (unit) (x_3_)	[27,30]
		number of A-grade tourist attractions (unit) (x_4_)	[27,28,32,34]
		Employees in the tourism industry (person) (x_5_)	[26,27,28,32]
		Original value of the tourism enterprise fixed assets (10,000 yuan) (x_6_)	[27,28]
	Industrial performance	International tourism receipts (10,000 US $) (x_7_)	[25,26,32,33,34]
		Number of international tourists (millions) (x_8_)	[25,26,32,33,34]
		The average daily per capita expenditure by international tourists (US $/daily per capita) (x_9_)	[33]
		Room occupancy rate (%) (x_10_)	[27,28]
		Tourism operating receipts (10,000 CNY) (x_11_)	[38]
Transportation subsystem (y)	Industrial scale	Length of highways in operation (km) (y_1_)	[25,33]
		Length of railways in operation (km) (y_2_)	[33]
		Number of civilian passenger cars (10,000 units) (y_3_)	[25]
		Municipal area of paved roads (10,000 sq.m) (y_4_)	[25]
	Industrial performance	Railway passenger traffic (10,000 persons) (y_5_)	[25,32,33]
		Highway passenger traffic (10,000 persons) (y_6_)	[25,32,33]
		Civil aviation passenger traffic (10,000 persons) (y_7_)	[25,33]
		Railway passenger turnover (100 million per-km) (y_8_)	[25,32,33]
		Highway passenger turnover (100 million per-km) (y_9_)	[25,32,33]
		Civil aviation passenger turnover (100 million per-km) (y_10_)	[25,33]
Low-carbon city subsystem (z)	Energy consumption	Total energy consumption (10,000 tonnes of SCE) (z_1_)	[27]
		Energy consumption per unit of GDP (tonnes of SCE/10,000 CNY) (z_2_)	[27,31,34,35]
		Annual per capita energy consumption (kg of SCE) (z_3_)	[34]
		Energy consumption elasticity ratio (z_4_)	[27,34]
	Carbon emission	Carbon emissions per capita (tonnes/person) (z_5_)	[23,27,31,34,37]
		Total carbon emissions (10,000 tonnes) (z_6_)	[23,27]
		Carbon productivity (10,000 CNY/10,000 tonnes) (z_7_)	[24,27,34]
	Low-carbon environment	Innocuous disposal rate of living garbage (%) (z_8_)	[27,34]
		Afforestation area per capita (1000 Ha) (z_9_)	[39]
		Urban green coverage rate (%) (z_10_)	[31,34]

**Table 2 ijerph-17-00792-t002:** Discriminating standard for the coupling coordination degree.

Range	Scoring Standard	Classification
Coordinated development (acceptable)	0.8 < D < 1	High coordination
	0.7 < D < 0.8	Intermediate coordination
	0.6 < D < 0.7	Primary coordination
Transitional development	0.5 < D < 0.6	Reluctant coordination
	0.4 < D < 0.5	Approaching imbalance
Imbalanced development (unacceptable)	0.3 < D < 0.4	Slight imbalance
	0.2 < D < 0.3	Moderate imbalance
	0 < D < 0.2	High imbalance

**Table 3 ijerph-17-00792-t003:** Index weights for the three subsystems.

Subsystem	First-Class Index	Weight	Second-Class Index	Weight
Tourism subsystem (x)	Industrial scale	0.4733	Total star-rated hotels (number) (x_1_)	0.0603
			Total travel agencies (number) (x_2_)	0.0746
			Total number of beds in hotels (unit) (x_3_)	0.0936
			number of A-grade tourist attractions (unit) (x_4_)	0.1339
			Employees in the tourism industry (person) (x_5_)	0.0511
			Original value of the tourism enterprise fixed assets (10,000 yuan) (x_6_)	0.0598
	Industrial performance	0.5267	International tourism receipts (10,000 US $) (x_7_)	0.0914
			Number of international tourists (millions) (x_8_)	0.0976
			The average daily per capita expenditure by international tourists (US $/daily per capita) (x_9_)	0.1521
			Room occupancy rate (%) (x_10_)	0.1010
			Tourism operating receipts (10,000 CNY) (x_11_)	0.0846
Transportation subsystem (y)	Industrial scale	0.3637	Length of highways in operation (km) (y_1_)	0.0844
			Length of railways in operation (km) (y_2_)	0.1135
			Number of civilian passenger cars (10,000 units) (y_3_)	0.0959
			Municipal area of paved roads (10,000 sq.m) (y_4_)	0.0699
	Industrial performance	0.6364	Railway passenger traffic (10,000 persons) (y_5_)	0.1264
			Highway passenger traffic (10,000 persons) (y_6_)	0.1125
			Civil aviation passenger traffic (10,000 persons) (y_7_)	0.0886
			Railway passenger turnover (100 million per-km) (y_8_)	0.1118
			Highway passenger turnover (100 million per-km) (y_9_)	0.0994
			Civil aviation passenger turnover (100 million per-km) (y_10_)	0.0977
Low-carbon city subsystem (z)	Energy consumption	0.4987	Total energy consumption (10,000 tonnes of SCE) (z_1_)	0.1479
			Energy consumption per unit of GDP (tonnes of SCE/10,000 CNY) (z_2_)	0.0967
			Annual per capita energy consumption (kg of SCE) (z_3_)	0.1462
			Energy consumption elasticity ratio (z_4_)	0.1079
	Carbon emission	0.3188	Carbon emissions per capita (tonnes/person) (z_5_)	0.0715
			Total carbon emissions (10,000 tonnes) (z_6_)	0.0773
			Carbon productivity (10,000 CNY/10,000 tonnes) (z_7_)	0.1702
	Low-carbon environment	0.1825	Innocuous disposal rate of living garbage (%) (z_8_)	0.0459
			Afforestation area per capita (1000 Ha) (z_9_)	0.0818
			Urban green coverage rate (%) (z_10_)	0.0548

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
