# Peer review of "Tourism, Transportation and Low-Carbon City System Coupling Coordination Degree: A Case Study in Chongqing Municipality, China"

_ijerph, 2020, doi:10.3390/ijerph17030792_

Round 1
Reviewer 1 Report
In this manuscript, the authors investigate the relation between tourism, transportation and low carbon cities.
They use EWM, Gray relation analysis and CCDM. All methods are sufficiently explained, however there are still some questions that should be answered.
The choice of the indices seems intuitive for Tourism and Transport. However, one problem is that for the chosen approach all indices are weighted and added together. This is problematic for indices like occupancy rate, which more or less should be multiplied with the other indicators. (Number of bed times occupancy is a better indicator than alpha * number of beds + beta * occupancy.) The authors could argue, why their approach is still valid, or multiply the relative indices beforehand, so that the weighting still works.
For low-carbon, the choice of the indices seems less intuitive, especially the subindices of low-carbon environment. The author could explain better, why they think that those indices describe the system best. Also they should explain why the carbon emissions are later not part of the low-carbon index, but plotted separately. Would the result change drastically, if the emissions would be included?
Figure 3 is a bit confusing. From the explanation in the text, I would expect to see a value for the tourism subsystem, and one for each subsystem. The whole system should then be a (weighted) sum of the two subsystems. However, in the plot the whole system level is below both subsystems. What does this mean? (same for Figure 4)
A final remark on the interpretation of the results: I think it is very important to mention that the authors observed a correlation, but no causal relation. This means we cannot make predictions about the behavior of one system, when we change another system.
Author Response
Dear reviewer,
Thank you for your comments concerning our manuscript. Those comments are all valuable and very helpful for revising and improving our paper, as well as the important guiding significance to our research. We have studied comments carefully and have made correction which we hope meet with approval.
We have made changes for the manuscript, as shown in the updated version, and the following are reply to the reviewer point to point.
Point 1: Index system: One problem is that for the chosen approach all indices are weighted and added together. This is problematic for indices like occupancy rate, which more or less should be multiplied with the other indicators. (Number of bed times occupancy is a better indicator than alpha * number of beds + beta * occupancy.)
Response 1: Room occupancy rate refers to the number of rooms (nights) actually sold divided by the number of rooms (nights) available within the reporting time frame, which can well represent the hotel’s operating conditions and can also reflect the performance of the tourism industry to a certain extent. In the processing of this indicator, this article refers to some articles. In the processing of this index, this article refers to articles 1-3. So we think our approach is valid.
Zhang, H.; Gu, C. L.; Gu, L. W. The evaluation of tourism destination competitiveness by TOPSIS & information entropy – A case in the Yangtze River Delta of China. Tourism Management 2011, 32(2): 443-451. Liang, X. D.; Liu, C.M.; Li, Z. Measurement of Scenic Spots Sustainable Capacity Based on PCA-Entropy TOPSIS: A Case Study from 30 Provinces, China. International Journal of Environmental Research and Public Health 2017, 15(1):10-. Wang, Q. R.; Mao, Z. X.; Xian, L. H.; Liang, Z. X. A study on the coupling coordination between tourism and the low-carbon city. Asia Pacific Journal of Tourism Research 2019, 24(6): 550-562.Point 2: Index system: For low-carbon, the choice of the indices seems less intuitive, especially the subindices of low-carbon environment. The author could explain better, why they think that those indices describe the system best. Also they should explain why the carbon emissions are later not part of the low-carbon index, but plotted separately. Would the result change drastically, if the emissions would be included?
Response 2: Thank you for your reminder. Based on your comments, For low-carbon city subsystems, we have added explanations for the index at all levels. For details, please see the index system section of our revised edition in lines 187-208. Also, we have revised the statement about total carbon emissions (formerly CO2 emissions inventory). Total carbon emissions (Second-class index) is included in the first-class index of carbon emissions (Table 1).
Point 3: Results: Figure 3 is a bit confusing. From the explanation in the text, I would expect to see a value for the tourism subsystem, and one for each subsystem. The whole system should then be a (weighted) sum of the two subsystems. However, in the plot the whole system level is below both subsystems. What does this mean? (same for Figure 4)
Response 3: Thank you for your valuable recommendations. According to your suggestions, we found that we had made mistakes in calculating the industrial scale and industrial performance level in the tourism subsystem, as well as in the transportation subsystem and the low-carbon city subsystem. We have used the grey relational model to calculate correctly again and modified Figure 3, Figure 4 and Figure 5 and the corresponding result analysis. And now the whole system is a (weighted) sum of the two/three subsystems.
Point 4: Results: I think it is very important to mention that the authors observed a correlation, but no causal relation. This means we cannot make predictions about the behavior of one system, when we change another system.
Response 4: Thank you for your valuable feedback. Based on your suggestions, In the results section we rearranged the functional relationship between tourism, transportation and low-carbon city subsystem, looking for internal causal relationship. And the result analysis of the coupling coordination of the three subsystems has been revised in the manuscript. Thanks again for your guidance.
We tried our best to improve the manuscript and made some changes in the manuscript.
Once again, we appreciate for reviewers’ warm work earnestly, and hope that the revised version will meet with approval.
Zhi Li
Sichuan University
2020.1.23

Reviewer 2 Report
Dear colleagues,
The research focuses on the effect the tourism industry and the transportation system have on the environment and the ways of implementing sustainable tourism development in big cities. The authors provides a sound scientific basis to prove close correlation among tourism, transportation and developing low-carbon cities.
Another point that gives additional advantage to the paper is that the researchers build their analysis on the data collected for the period covering the years 2008-2018, which adds validity to the outcomes they arrive at. Up to the merits of the research is the fact that the authors give a detailed and sound description and explanation of the survey tools they have employed in their work. All this mentioned above well substantiates the outcomes of the research and authors' conclusions.
The only point that I would like to draw the authors' attention to is a kind of arithmetical flaw in lines 154-156: "The area under the Chongqing jurisdiction is 470 kilometers wide from east to west and 450 kilometers long from north to south; a total area of 82.40 square kilometers." - When multiplying 450 km and 470 km I fail to come to 82.40 km2 . If it is a mistake it may be easily corrected.
In my view the article is complete, original, scientifically sound. Besides the analysis and well-developed conclusions it comprises some recommendations for improving the situation, which adds some extra value to the work. The research as it is should be of interest both to other colleagues and to professionals in the relevant spheres and can be published.
Kind regards
Author Response
Dear reviewer,
Thank you for your comments concerning our manuscript. Those comments are all valuable and very helpful for revising and improving our paper, as well as the important guiding significance to our research. We have studied comments carefully and have made correction which we hope meet with approval.
We have made changes for the manuscript, as shown in the updated version, and the following are reply to the reviewer point to point.
Point 1: Study area: The only point that I would like to draw the authors' attention to is a kind of arithmetical flaw in lines 154-156: "The area under the Chongqing jurisdiction is 470 kilometers wide from east to west and 450 kilometers long from north to south; a total area of 82.40 square kilometers." - When multiplying 450 km and 470 km. I fail to come to 82.40 km2. If it is a mistake it may be easily corrected.
Response 1: Thanks for your comment. Based on your comments, we have checked relevant information: the area of Chongqing is 8.24 square kilometers in lines 155-156. It has been revised.
We tried our best to improve the manuscript and made some changes in the manuscript.
Once again, we appreciate for reviewers’ warm work earnestly, and hope that the revised version will meet with approval.
Zhi Li
Sichuan University
2020.1.23

Round 2
Reviewer 1 Report
Thank you for this revision.
All raised points have been addressed sufficiently.